# Large-Scale Mobile-Based Analysis for National Travel Demand Modeling

Bat-hen Nahmias-Biran [1,2,*], Shuki Cohen [3], Vladimir Simon [4] and Israel Feldman [5]

1 Department of Civil and Environmental Engineering, Ariel University, Ariel 40700, Israel
2 Department of Civil and Environmental Engineering, Massachusetts Institute of Technology, Cambridge, MA 02139, USA
3 MATAT—Transportation Planning Center Ltd., Azor 58190, Israel
4 Ministry of Transport and Road Safety, Israel Bank 5, Jerusalem 9195021, Israel
5 Mobility Insight Ltd., Hatahana 1, Kfar Saba 4418001, Israel
* Correspondence: bathennb@ariel.ac.il

**Abstract:** Mobile phones have achieved a high rate of penetration and gained great interest in the field of travel behavior studies. However, mobile phone data exploitation for national travel models has only been sporadically studied thus far. This work focuses on one of the most extensive cellular surveys of its kind carried out thus far in the world, which was performed for two years between 2018 and 2019 with the participation of the two largest cellular providers in Israel, as well as leading GPS companies. The large-scale cell phone survey covered half the population using cellphones aged 8+ in Israel and uncovered local and national trip patterns, revealing the structure of nationwide travel demand. The methodology consists of the following steps: (1) plausibility and quality checks for the data of the mobile operators and the GPS data providers; (2) algorithm development for trip detection, home/work location detection, location and time accuracy, and expansion factors; (3) accuracy test of origin–destination matrices at different resolutions, revisions of algorithms, and reproduction of data; and (4) validation of results by comparison to reliable external data sources. The results are characterized by high accuracy and representativeness of demand and indicate a strong correlation between the cellular survey and other reliable sources.

**Keywords:** travel demand; cellular data; mobile phone data; travel surveys; national model

## 1. Introduction

Transport planners mostly rely on transport demand models for the understanding of mobility behavior and the planning of network infrastructure [1]. Transport travel demand models heavily rely on high-cost and hard-to-update travel surveys as a data source and thus cannot be updated regularly. Furthermore, travel habit surveys provide detailed and in-depth information on the travel habits of those sampled for the survey; these surveys are conducted at the city or region level, with other, lower scope surveys not allowing the required information on the national travel system to be generated with appropriate resolution. Additionally, traditional collection methods result in an overview of the mobility of one weekday; therefore, they only provide a snapshot of people's movement since they cover a limited sample of the population and a small time window. As a result, travel demand models may not reflect the variability in travel and travel changes over time. Therefore, new data sources that are richer and more available are needed [2]. Although there are studies analyzing mobile phone data to understand mobility patterns, they often rely on single days, a very small sample, and are limited in terms of coverage area. Therefore, the reliability of the results is often questionable [3]. Moreover, mobile phone data exploitation for national travel models has only been sporadically studied thus far [4]. This work focuses on one of the most extensive cellular surveys of its kind carried out thus far in the world.

Cell phone data were collected in Israel for two years in 2018 and 2019 and recorded approximately 2.67 billion person days and approximately 4.6 billion trips of 1.5 km or more. The information collected was not based on a sample but on all of the subscribers of the largest cell phone operators in Israel, Cellcom and Pelephone. The data collected grouped together approximately 3.7 million active users, which are approximately 50% of the relevant population in the State of Israel (it can be assumed that the users are aged 8+). The cellular providers were supplemented by the leading GPS companies "Pango", "Anagog", and "Neura", which collected data from cellular apps based on GPS locations for the purpose of validation and for special processing. Both companies recorded hundreds of location data points on an average day per subscriber [5].

The extraction of transportation data from cell phone observations is a complex process and is accompanied by many challenges: from trip identification, home and workplace/study location identification, accuracy in associating origin and destination with traffic analysis zones (TAZ), identification of travel route and user, and travel time calculation accuracy to producing origin–destination matrices with the required resolution given the privacy limitations. Thus, this research investigates whether it is possible to produce detailed origin–destination matrices from cellular data (CD) and to what level of accuracy it can be achieved. Are mobile data capable of replacing traditional data collection methods, and what are the implications of using cellular data for demand modeling and for estimating a national travel model?

In a classical four-step model, the origin–destination matrices are yielded in two steps: (1) the trip generations/attractions are modeled, and (2) the origin–destination matrices are acquired according to these generations/attractions and based on travel impedances [1]. However, the methodology developed in this paper removes these two steps and, more importantly, provides data that can be updated regularly. In this way, the need for estimating regression models for generations/attractions as well as the calibration of gravity-type models are omitted. Indeed, the results indicate a strong correlation between the CD and other reliable external sources, such as the Global Positioning System (GPS), Israel's Central Bureau of Statistics (CBS), travel habit surveys (THS), and traffic monitoring surveys.

This paper is organized as follows: Section 2 provides a literature review regarding the use of mobile phone data for human mobility research focusing on origin–destination matrix extraction for city, regional, and national demand modeling. Section 3 introduces the data sources and methodology used in this research, where algorithms and quality checks are discussed. Section 4 includes the results and validation of origin–destination matrices at different resolutions using multiple data sources. Finally, Section 5 presents the main conclusions and findings of this work.

## 2. Related Work

The wide adoption of mobile devices and the rapid related advancements make mobile phone data especially suitable for the study of human mobility for transportation research. Mobile phones have achieved a high rate of penetration and gained great interest in the field of travel behavior studies. In Israel, the percentage of smartphone ownership is one of the highest in the world, and approximately more than nine in ten Israelis own smartphones [6]. Mobile phone data have been explored for mobility pattern extraction [7–11], traffic and mobility flow inference [9,12–16], population estimation [17–19], and route choice modeling [19]. Moreover, mobile phone signaling data have been explored to detect individual activities and activity plans [20–25] and to infer travel modes [26–29]. Furthermore, cell network traces were used to extract transport-related measures such as the mean speeds, travel distance, and journey times [4,8,12].

Early studies tried to extract O-D matrices based on very small samples of CD covering a very limited area [11,30]. Later, [12] produced an O-D matrix from a detailed mobile phone dataset for the Boston region in Massachusetts, comparing it with O-D flows from census data taking into account only weekday morning trips. In [31], an algorithm method was adapted to the available database of mobile device records that covers a large territory

of Sweden to generate O-D flows. However, no detailed comparison for the entire matrix was performed in this work. In [32], "transient O-D matrices" were generated and converted into intersection-to-intersection O-D flows in the road network of Boston and San Francisco. To calibrate the derived O-D trips, the authors used available travel data, vehicle usage rates, and population statistics. The same was carried out for Dhaka, Bangladesh in [33]. The authors used limited traffic counts and a microscopic simulation, scaled up the generated OD patterns (from calling data) and validated the assignment results with additional traffic counts. In [34], an analysis on triangulated CD data was conducted to infer O-D individual trips per purpose (home, work, or other) and time of day. Using a very small sample, the authors validated the results against travel surveys and census data on the Boston metropolitan area. In addition, based on CD datasets provided from Ivory Coast and Senegal territories, flow estimation for mobility metrics extraction was explored in [35]. However, no validation of this analysis was performed due to lack of data. To increase sample reliability, some studies combined CD data with other urban transportation data sources, such as GPS data (e.g., from taxis, private cars, or mobile phone applications) [14,36,37], smart-card data [14], travel surveys, and existing transport models [15,38]. More recent work [39] used signaling data collected from the 2G network in 2009 to produce the O-D matrix of individual travel and compared them with the local household travel survey in the Paris region. The authors obtained similar estimations for O-D pairs with high traffic. The same form of data was analyzed in [40] in Hangzhou (China). In [14], a data-driven real-time mobility model for the city of Shenzhen (China) was proposed that combines the advantages of 2G mobile phone signaling records (of one day) and urban transportation data. The model validation was performed by comparing the predicted mobility flows and the travel demands obtained from the same signaling data used to build the model, as no other data were available for evaluation.

In [9], a methodology is proposed to estimate O-D matrices based on s from 2G and 3G cellular network signaling data in the Rhône-Alpes region, France. However, only a fraction of the population was observed for a 24 h period. Even so, this methodology can perform scaling and shows that a signaling data-based O-D matrix carries similar estimations as those that can be obtained via travel surveys.

None of the previous works appear to have achieved reliable complete O-D matrices using only signaling data apart from [9]. Moreover, only a few studies have addressed the validation of the outcome and the accuracy of results, and some of them have used the same data for the matrix estimation and validation [41,42]. Additionally, in numerous studies, scholars have evaluated only travel flow structure and trip distribution instead of trip volumes [38,43] since adequate methods to expand inferred O-D matrices are still missing to characterize the whole population.

Given the challenges for a small- and medium-sized scale, mobile phone data exploitation for national travel models has only been sporadically studied thus far. In [44], mobile phone data and national travel survey results in Israel were compared. However, the study sample was comparatively small in terms of person days. Similar work has been done in the USA (North Carolina) [45] and in France [9] but was also limited to a regional level. Recently, [30] analyzed cellular data collected over five months, covering 35.9% of the French population. However, the authors based their work on a low frequency of cellular data from 2007, which is likely to be responsible for a substantial underestimation of the long-distance tour rate that they examined.

The research described herein thus closes the gaps found in the literature, analyzing and validating the most extensive cell phone survey carried out thus far in the world at the national level for the purpose of revealing the structure of nationwide travel demand.

## 3. Data and Methods

The methodology consists of the following steps: (1) plausibility and quality checks for the data of the mobile operators and the GPS data providers; (2) algorithm development for trip detection, home/work location detection, location and time accuracy, and expansion

factors; (3) accuracy test of origin–destination matrices at different resolutions, revisions of algorithms, and reproduction of data; and (4) validation of results by comparison to reliable external data sources (such as CBS data and THS data). These steps are described and discussed in detail in this section.

The basic information collected by the cell data providers is a series of basic location points (cell points) with an accuracy of hundreds of meters and the time at which the cell phone was registered in the cell point. On the basis of this series of points, the stop zones within the TAZ were identified. By using appropriate algorithms, "travel diaries" are generated for each user, through which the origin–destination (O-D) matrices are generated.

The process of identifying trips starts by identifying intermediate points (way points) where the user stays beyond a defined time (several minutes) in a limited area, which is expressed by a jump between near basic locations. With the identified intermediate points, using an additional algorithm, stop points constitute activity destinations on the basis of which the origin and travel destinations (trips) are determined. In addition, in terms of user intermediate points, over time, significant locations can be identified for each of the users, such as the location of the user's home and work zones. Identifying these locations was an important part of the stop detection process, as well as significant information for transportation analyzers.

After generating a travel diary for each of the users, travel matrices can be generated. The main challenge at this stage is considering privacy regulations, according to which it is not possible to export zone-to-zone pair information where the sum of trips is less than 50. For this purpose, algorithms were developed based on aggregation in time and space, which allowed the creation of detailed matrices, including pairs, with few trips made throughout the period. This section describes the process of transforming the cellular observations into transport data, which are eventually generated into O-D matrices for the purpose of estimating the demand nationally, with an emphasis on the algorithms and logic tests used in this process.

### 3.1. Data Sources

The research described in this paper is based on cellular data (CD) that were collected for two years in 2018 and 2019. The information collected was not based on a sample but on all of the subscribers of the largest cell phone operators in Israel, Cellcom and Pelephone. Table 1 shows a comparison between cell phone operator characteristics.

**Table 1.** Comparison of Cellcom and Pelephone characteristics.

| Characteristic | Cellcom | Pelephone |
|---|---|---|
| Number of subscriptions (monthly average) | 1.9 million Israeli subscribers and 92,100 tourists. | 1.76 million Israeli subscribers and ~100k tourists. |
| Density of monitoring per subscriber (quantity of records) | 720 billion records per year, 918 daily data per subscriber, every 94 s. | 929 billion records per year. 1000 daily data per subscriber, 22% more data on weekdays than on weekends. |
| Record details (raw data) | Each record includes a location and a time tag. | Each record includes a location and a time tag. |
| Position registration frequency | 2 min on average. A rate of 30 records per minute during calling time. | Every 30 min if not active, 2 min on average. |

A preliminary experiment aimed at examining the degree of continuity and accuracy of travel monitoring by cellular companies was carried out over a period of three weeks. A double-tracking experiment was carried out using 140 volunteers selected by the Israeli MoT and cellular companies. At the same time, the registration of cell phone locations (CP) was made according to the GPS app installed on volunteer phones. This app, developed for the purpose of the experiment, records the locations of the cell phone throughout the day by Google's TimeLine application. Comparing the location and travel records, using both methods on the same phones, made it possible to determine the accuracy of the cell phone companies' reports. The accuracy of cell phone companies in locating trip

endpoints in relation to GPS monitoring was examined and is presented in Table 2. The results showed that the cellular providers detected 95% of the trips over 10 km. Below this distance, approximately 94% were located by the Pelephone network and 91% by the Cellcom network. The accuracy of the location was found to overlap with the antenna density, which was relatively high in built-up areas and lower in non-built-up areas. The inaccuracy in identifying the ends of the trip is approximately 300 m in built-up areas and approximately 850 m in non-built-up areas. The median and average error in the identification of place of residence and work was found to be lower.

**Table 2.** Experimental results comparing the two cellular providers.

| Comparison Parameters | | Pelephone | Cellcom |
|---|---|---|---|
| Built-up areas | Number of locations | 1317 | 673 |
| | Average distance relative to GPS (m) | 379 | 438 |
| | Median distance (m) | 251 | 336 |
| | Standard deviation (m) | 374 | 378 |
| | Average distance from/to GPS mapped trips * | 387 | 520 |
| Non-built-up areas | Number of locations | 120 | 121 |
| | Average distance relative to GPS (m) | 1202 | 913 |
| | Median distance (m) | 946 | 748 |
| | Standard deviation (m) | 728 | 666 |
| | Average distance from/to GPS mapped trips * | 1063 | 780 |

* Only accurately documented GPS trips based on distance between start and end of previous trip.

The accuracy of the cellular providers in detecting the locations of home and work in relation to GPS monitoring was also examined. The results show that the ability of the two suppliers is similar in terms of the quality of the data they provided. The level of geographic accuracy was generally high, especially in identifying the home and work addresses. However, the analysis of the cellular operator results and the manner in which these results were obtained indicated that there are differences in working methods, algorithms used, the layout of the cellular network, and the characteristics of the subscriber population to some extent. Nevertheless, the experiment revealed two vulnerabilities: (1) a relatively low detection rate for short trips and (2) inaccurate timing of the start and end of trip detection. Therefore, it was decided in the full analysis to filter trips shorter than 1.5 km and to perform strict quality control for the results of trips shorter than 5 km. It was further decided to conduct a methodological regulation that would ensure uniformity in the algorithms and results. In addition, it was decided that both companies should do the collection and processing of the data throughout the data collection period simultaneously. Equally important are the representations of the studied population due to the difference in the subscriber population of each of the companies. The remainder of this section addresses the methods developed for the processing of the cellular data as part of the methodological regulation that was used in this work by both cellular providers.

### 3.2. Trip Detection

Cellular provider data were processed to identify trips to estimate the number of hourly trips in Israel for all trip purposes at all required resolutions. The process required (a) identification of travel, (b) identification of home–work locations, and (c) ensuring the accuracy of the location, as described in the following paragraphs.

### 3.2.1. Trip Definition

The definition of "trip" underwent several rounds of improvements until a series of tests were set to identify "trip", which were then implemented in the processing procedure of the raw data by the cellular providers. As part of this processing procedure, the sequence of basic location points of the user (Cell Points) were scanned, and a set of adjacent points in the space identified that the user passes for an extended period of time, as a static cell phone

typically tends to bounce between nearby antennas. Each such set of points identified is defined as a way point, and the centroid position of that set is calculated as the reference of that point. Staying at way points for a long period ends the journey and defines the "trip". Thus, the initial definition of a trip used by the cellular providers was that a stop at a way point over a set time (15 min by Cellcom and 20 min by Pelephone) ends a trip. Apart from the non-uniformity in time set between the providers, there were at least two problems with such a definition. The first is incorrect trip split. For long trips, a stop for rest, refueling, etc., is possible for more than 15–20 min, followed by the continuation of the trip afterwards. In such a case, the trip may be displayed as two trips, misrepresenting the origin and destination. The second problem is incorrect consolidation. On short trips, one can take a trip and travel to a different purpose and destination in less than 15–20 min. In such a case, the 15–20 min stop criterion, as a minimum to define the end of a trip, consolidates two different, short trips incorrectly as a single longer trip. A typical example of this is the movement of a commercial vehicle loading/unloading goods at stops of less than 15–20 min. To avoid these and other errors, more complex criteria for the accurate diagnosis of trips based on the received data were developed using the following:

- A stop of less than 8 min will always be considered a way point and not an end trip. This parameter represents the minimal time to stop a trip, in the case of home or work trips. The data showed that a time shorter than 8 min can be interpreted as waiting at a way point for any reason but is not a real stop. A deliberate stop for the purpose of home or work will include the time of entering the building (2–3 min) plus at least 5 min of staying in it, so it can be said that the stop is a "real stop".
- A stop of more than 40 min will always be considered the end of a trip. This parameter was chosen by the Israeli MoT as the ultimate trip terminator, as the data showed that, for long trips, a stop of 30–40 min is customary for a food break or refreshment; beyond that, it is assumed that other activities are combined, so it is not a break but a "real stop".
- A stop over 8 min will always be considered a new trip if the stop is the person's home or workplace.
- For stops that are not home or workplace and that take place between 8 and 40 min:
  - If the distance from the origin to destination is shortened after the stop, it is a new trip.
  - If the distance from the origin to destination is increasing.
  - If the duration of the stop is less than half the travel time until the stop, it is the continuation of the trip and not a new trip.
  - If the duration of the stop is longer than half the travel time until the stop, it is a new trip.

Only trips that were 1.5 km or longer were included in this analysis. Although this definition turned out to be accurate in most cases, there may still be instances where the trip distinction was incorrect; for example, a 7 min stop not at home or in the workplace (e.g., near a kindergarten) and a trip back to the origin point. It is assumed that these situations are not common in trips over 1.5 km in length. Figure 1 shows the described trip detection mechanism.

### 3.2.2. "Home" and "Work/Study" Detection

Work–home relationships constitute key information for transportation modelers. The original (of the cellular providers) definition of "home" and "work" locations included two criteria: (a) "home" was calculated and defined as the exact location of the subscriber's cell phone at night, while "work"/"study" was determined as the exact location of the phone during the day; (b) location of the "home" and "work"/"study" zone was determined on a monthly basis from the raw data. These definitions were found to be incomplete, thus, a revised definition was drawn up as follows:

- "Home" and "work"/"study" will be identified in the largest resolution of 2640 statistical zones. In addition, all the statistical zones enveloping the "home" and "work"/"study" zones will also be included. This is for cases where a subscriber park (or for other reasons) is in the vicinity of the "home"/"work"/"study" and not adjacent to it.
- The detection of "home" and "work"/"study" zones must stem from the trip diary as follows:

  ○ The use of two weeks of trip diary.
  ○ Trip diary must include the "home" and "work"/"study" zones, if any.
  ○ Trip diary is generated using algorithms for identifying a user's stops and way points throughout the day.
  ○ When the trip diary does not include the "home" or "work" zone or both, these cases are presumed to be errors.

- The minimum monthly stay in the "home" zone is 160 h during the month.
- The minimum distance between "home" and "work"/"study" is 1.5 km.
- The "home" location is a zone that attracts or generates most of the trips on weekdays.
- A "work"/"study" zone will be defined as one that is not a "home" and in which the user spends the maximum time during the month with stays over 3 h.
- The "home"–"work"/"study" matrix will include users who take trips from "home" and who arrive at "work"/"study" at least 5 days per month.

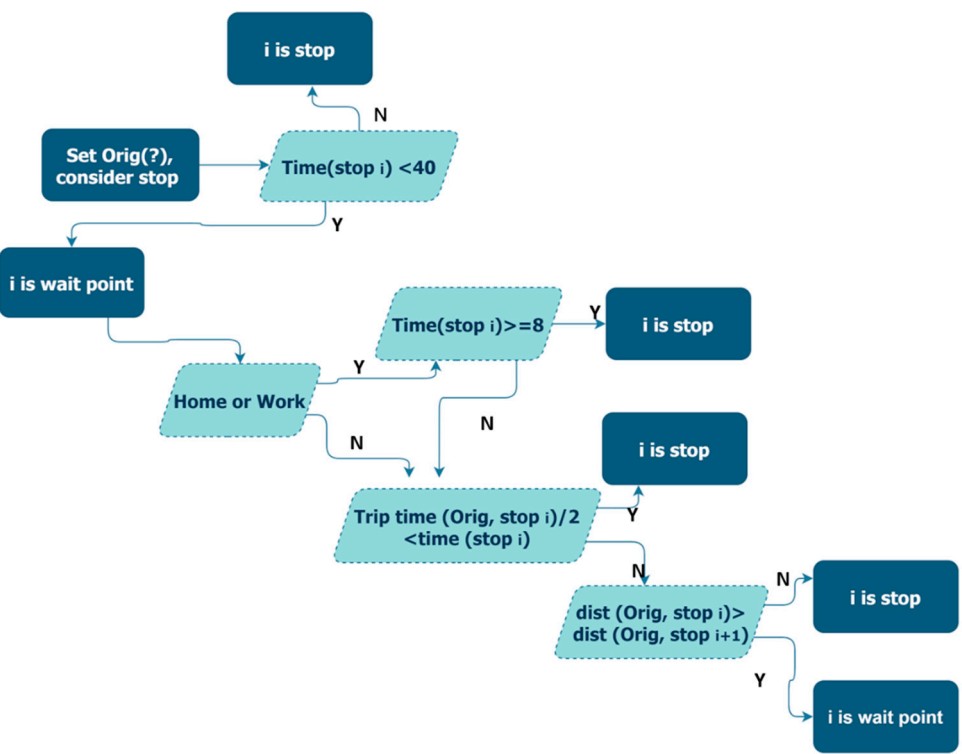

**Figure 1.** Trip detection mechanism.

3.2.3. Location Accuracy

The location of the user, O-D, and trip route are determined according to the deployment of the cellular antenna network and the algorithms that associate location data with the cellular phone that is being received in the network. The accuracy of the location affects the detection of the trip in terms of O-D zones and has other implications for trip characteristics. The precision limits of the cell phone were known in advance. In the preliminary experiment, it was found that the possible standard deviation of the location of the phone is up to approximately 400 m in a built-up area and up to approximately 700 m in a nonbuilt-up area. Such positioning accuracies do not allow the accurate association of the phone with "small" movement zones. For example, there are areas within the 2640 × 2640 zone

resolution that have an average size of $250 \times 250$ m$^2$ in built-up areas. In such cases, the chance of accurately connecting the phone to the zone is approximately 22%. Even for an area of $400 \times 400$ m$^2$, the chance increases but only to approximately 47%. Because of this, a person residing in a specific zone may be associated with a neighboring zone, or a trip to work that ends in an industrial area may be registered as ending in a different type of area. For this reason, the main resolution used in this work is 1270 zones. To this end, no technical solution was found for this limitation, and it was necessary to assemble a "logical control" to reduce/correct the distortions discovered. For this purpose, a control algorithm was developed. It is reasonable to assume that in the upcoming generation five and with more advanced mapping capabilities, the spatial positioning accuracy will improve.

*3.3. Expansion Factors*

Expansion factors were required to generate the O-D matrices for the full population of Israel. In this work, both data expansion and data weighting were considered part of the same process. Data expansion is simply the procedure of multiplying each observation in the data by a factor that represents how many members of the population are represented by that observation, as the data provided consisted of 50% of the relevant population in the State of Israel. Data weighting is the procedure of developing multiplication factors that attempt to correct for biases in the sample design that have been introduced, as each phone in each locality varies between cities, neighborhoods, and sectors. Throughout the cellular data collection period, the number of subscribers was similar between the two service providers, 1.3 million subscribers per day on average, but the distribution of subscribers in terms of regions and locales was different. The calculation of the expansion factors was done using TAZ resolution and in accordance with the following: let us define $P_i$ as the amount of population over 8 years of age in the locale sector (Arab, orthodox Jews, secular Jews) of $i$, $X_i$ is the amount of Cellcom subscribers living in locale sector $i$, and $Y_i$ is the amount of Pelephone subscribers living in locale sector $i$. Thus, each phone $F_i$ in each locality $i$ can be represented by:

$$F_i = P_i / (X_i + Y_i) \tag{1}$$

$F_i$ varies between cities/neighborhoods/sectors and was recalculated every month. The phone will count as $F$ phones during the entire month and on all trips, regardless of its location. The weight of Cellcom among the total number of phones of both companies is W and is defined as follows:

$$W = \sum_1^N X_i / \left(\sum_1^N X_i + \sum_1^N Y_i\right) \tag{2}$$

Thus, Cellcom's expansion factors in the resolution of $2640 \times 2640$ zones that are included in locale sector $i$ are

$$CellcomF = F_i / w \tag{3}$$

Therefore, Pelephone's expansion factors in the resolution of $2640 \times 2640$ zones, which are included in sector $i$, are

$$PelephoneF = F_i / (1 - w) \tag{4}$$

Merging of O-D data of the two companies will be done by using a weighted average:

$$United\ W = w \cdot Cellcom + Pelephone(1 - w) \tag{5}$$

The weighted average ensures reference to the sum of phones and the expansion factors of the sum of phones. In other words, the population's trips are represented by the sum of the two cellular companies' users living in a given area multiplied by the extrapolation factor derived from the urban average by sector (and not by extrapolation calculated directly from that region). To reach the correct expansion factors, an iterative process was needed, and several rounds of data processing and analysis were performed, relying on

the population data and age distribution of the CBS. Figure 2 shows the comparison of the population over the age of 8 per locale as reported by CBS compared to the number of cellular subscribers in the same locale, which was found in the CD.

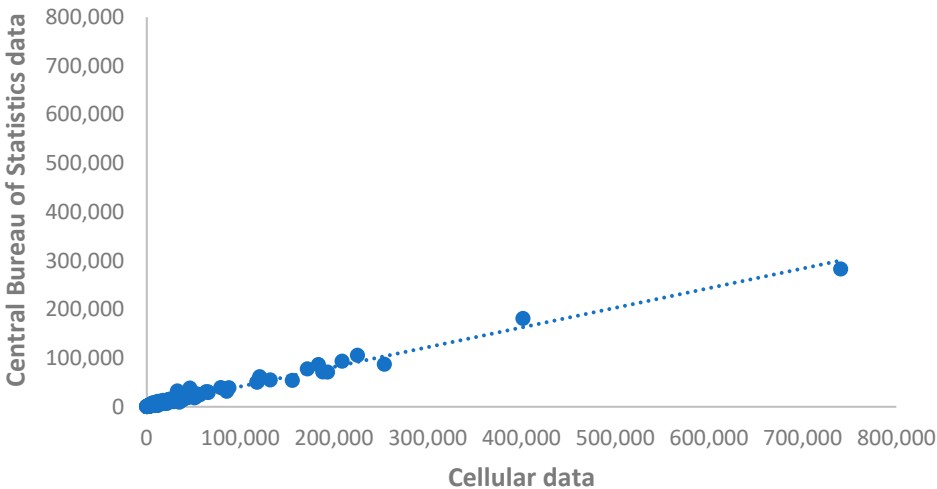

**Figure 2.** Population per locale: comparison of Central Bureau of Statistics data and cellular data.

The results show that a very high correlation coefficient of 0.982 was found between the total number of mobile phones of the two suppliers and the number of residents in a locale, taking into account the resident's sector. Considering the sector in determining the expansion factor is of great importance. For 80% of the population in Israel, the ratio between the number of people over the age of 8 in a locale and the sum of phones of Cellcom and Pelephone in that locale is greater than 40% and sometimes greater than 50%. Most of these are secular locales dispersed nationwide. However, there are quite a few locales where the ratio of the number of residents over the age of 8 in the locale to the number of Cellcom and Pelephone phones is significantly lower than 40%. The locales where the representation is relatively low are mainly Arab locales and orthodox locales or orthodox Jews areas within a secular locale. In addition, in those locales, there may be differences in population representation between Cellcom and Pelephone, as well as differences in trip preference to regions where the same sector lives. Expansion factors were verified by the Israeli Central Bureau of Statistics information, which provided data regarding the population and the number of Cellcom and Pelephone users in each settlement in Israel. In the CBS's list, it was possible to identify settlements that have a clear sectoral association-Arab settlements, ultra-Orthodox settlements, etc. Furthermore, in mixed cities, the CBS identified neighborhoods by sector and made it possible to perform validation by the research team.

*3.4. Privacy Challenges*

Cellular and GPS companies are required to comply with privacy limit rules, according to which it is not possible to report, in a separated point cell, fewer than 50 trips after expansion. This limitation does not allow a complete generation of O-D matrices at a resolution of 2640 statistical zones according to population sectors and/or age groups at an hourly time resolution. On average, approximately 95% of the point cells in the monthly O-D matrix at the national level do not meet the privacy limit prior to the segmentation that is required for the national model in age, sector, and modes of travel.

Figure 3 illustrates the matrix based on 33 statistical zones. Each color represents a statistical zone as a result of privacy limitations in the four largest metropolitan areas of Israel: Tel Aviv, Jerusalem, Haifa, and Beer Sheva. In accordance with Figure 3, Table 3 indicates the percentage of blank points as a result of privacy limitations. Since Figure 3 focuses on the largest metropolitan area in the state, where most of the trips are made, and due to the low matrix resolution (33 zones), most of the cells pass the privacy limitation.

However, considering trips between the remaining regions that are not part of the main metropolitan areas, the number of trips is small, and most of the point cells are blank.

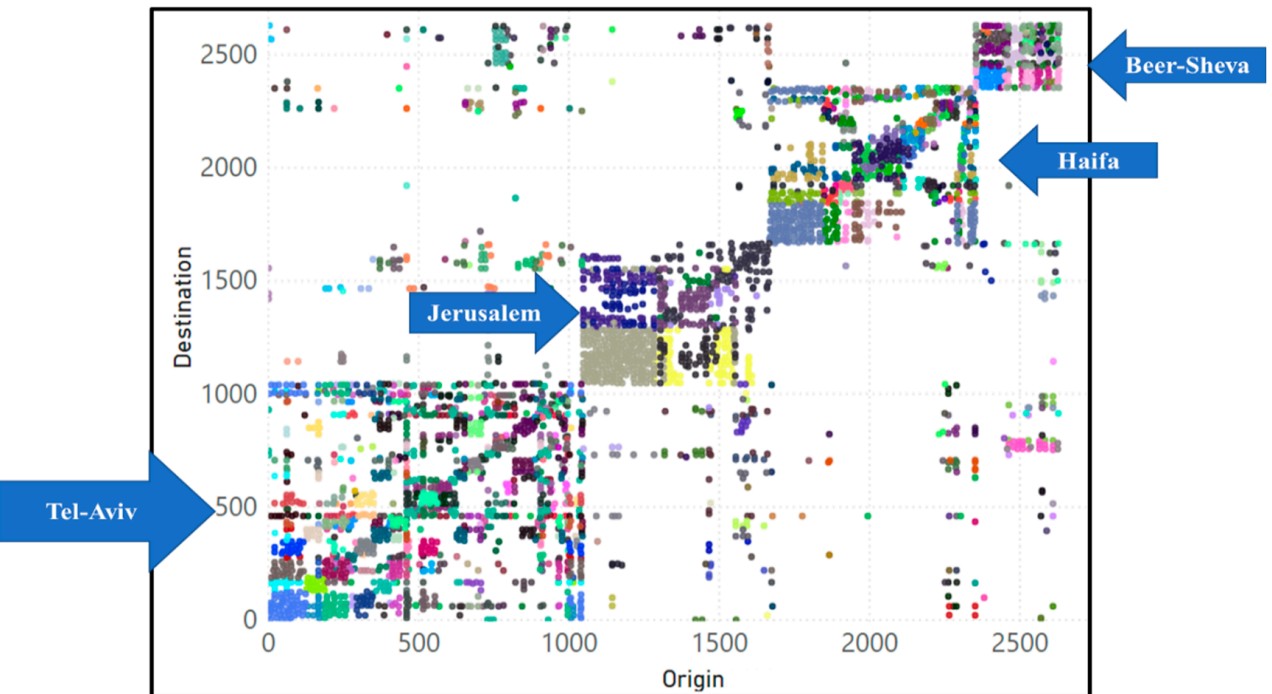

**Figure 3.** O-D blank point cell illustration as a result of privacy limitations.

**Table 3.** O-D blank point as a result of privacy limitations.

| Resolution | % of Passing | % of Not Passing | % of Blanks |
|---|---|---|---|
| 2640 × 2640 | 1 | 49 | 50 |
| 1270 × 1270 | 3 | 73 | 24 |
| 250 × 250 | 34 | 66 | 0 |
| 33 × 33 | 99 | 1 | 0 |

In view of the great difficulty that the privacy constraint posed, it was necessary to find a solution that allows us, on the one hand, to provide O-D matrices and, on the other, to maintain the privacy laws as needed. The solution found is based on aggregation and disaggregation in time and space and the accumulation of data over time. It is based on an algorithm that aggregates and disaggregates the zones and estimates the number of trips that did not exceed the privacy limit with an error of 20%. Specifically, solution implementation principles included the grouping of trip data over weekdays for 6 consecutive months. Then, instead of splitting a day into 24 periods of time, a day was defined as one period of time, and a distribution of the trips from origin to destination at the hourly level at a 1270 × 1270 resolution was calculated. This is done under the assumption that the hourly distribution of parent regions is not different from the distribution of child regions (2640 × 2640 resolution), which are similar in land use characteristics (residential, commercial, etc.). The implementation of this solution allowed the production of a complete daily trip O-D matrix (of weekdays) that includes 5.5 million O-D pairs. This was made possible due to the two-year data collection, which enabled the detection of infrequent/rare trips.

## 4. Results and Validation

The cellular data collection results included mainly O-D matrices varying in resolution, time section, and other transportation properties. Those were processed using two categories. The first is basic outputs, which included O-D travel matrices in different time sections and resolutions. Most of them are hourly matrices of 1270 by 1270 zones (by

weekday and weekend). Partially these include half-hour matrices or daily matrices with a multiplied resolution of 2640 by 2640 zones. The second is extended outputs, which included trip matrices sectioned by home–work association, commuting indicator, travel frequency, tour durations, and mobility of tourists. In this section, selected results will be shown.

### 4.1. Total Ridership and Distribution by Time of Day

In 2018 and 2019, an average of approximately 14.2 million rides, each longer than 1.5 km, was found in Israel using the CD. The daily peak hour was between 4 pm and 5 pm, averaging 1.04 million rides, 7.3% of total daily ridership. Slightly smaller numbers were observed between 7 am and 8 am. Figure 4a illustrates the trip distribution by day type and time of day, while the trip start time was measured. It was found that the share of night ridership is surprisingly high, and afternoon ridership is not significantly lower than during rush hour, but it is more dispersed. On Fridays, ridership sharply peaks between 11 am and 1 pm and between 12 pm and 1 pm, with more trips taking place than on weekdays. Saturday's peak is between 6 pm and 9 pm. The structure and distribution of travel length differs for each day type. Following traffic surveys conducted on monitored roads, early morning ridership (5 am to 6 am) on workdays has been growing significantly in recent years. Early morning hours generated approximately 1.84% of daily rides in 2018–2019 and were characterized by very long rides, as illustrated in Figure 4b.

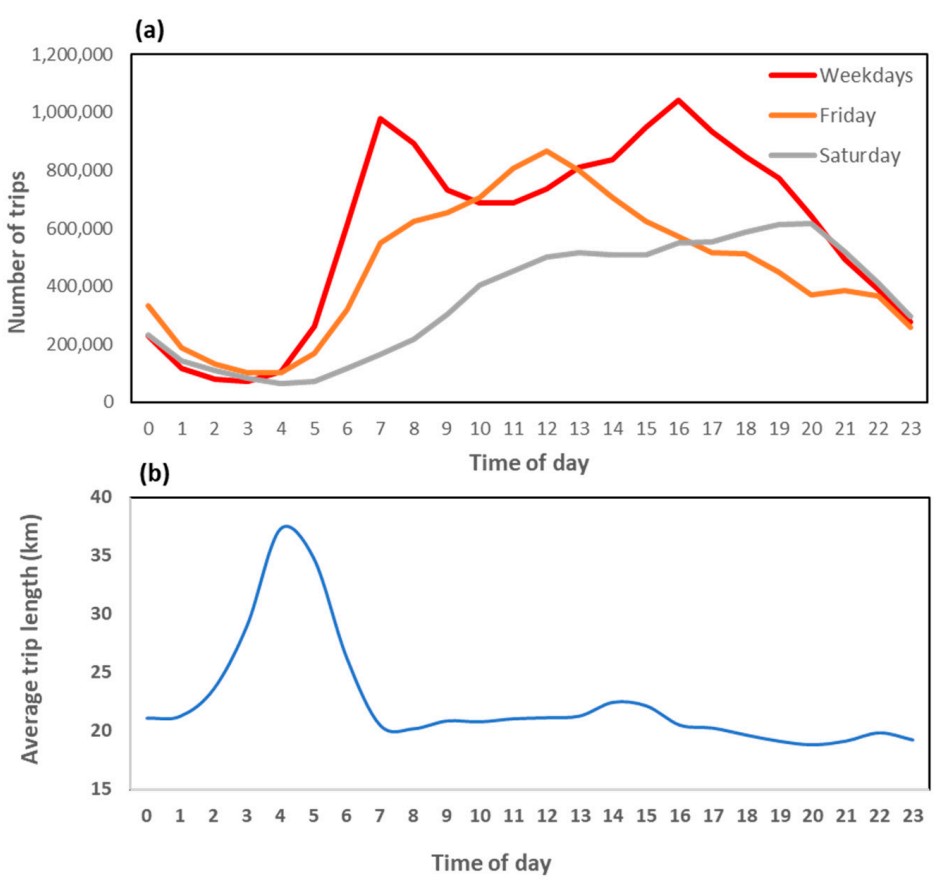

**Figure 4.** Trip distribution by (**a**) day type and time of day and (**b**) travel length and time of day.

Figure 5a,b shows incoming and outgoing trips for Tel Aviv and Jerusalem in a half-hour distribution, as well as trips taking place inside each of the cities. The ratio between incoming and outgoing rides in the morning for Tel Aviv is remarkable (almost 5:1 at 6 am); this ratio cuts in half at 8 am and turns into a surplus of outgoing ridership in the afternoons. The situation in Jerusalem is similar but not as pronounced.



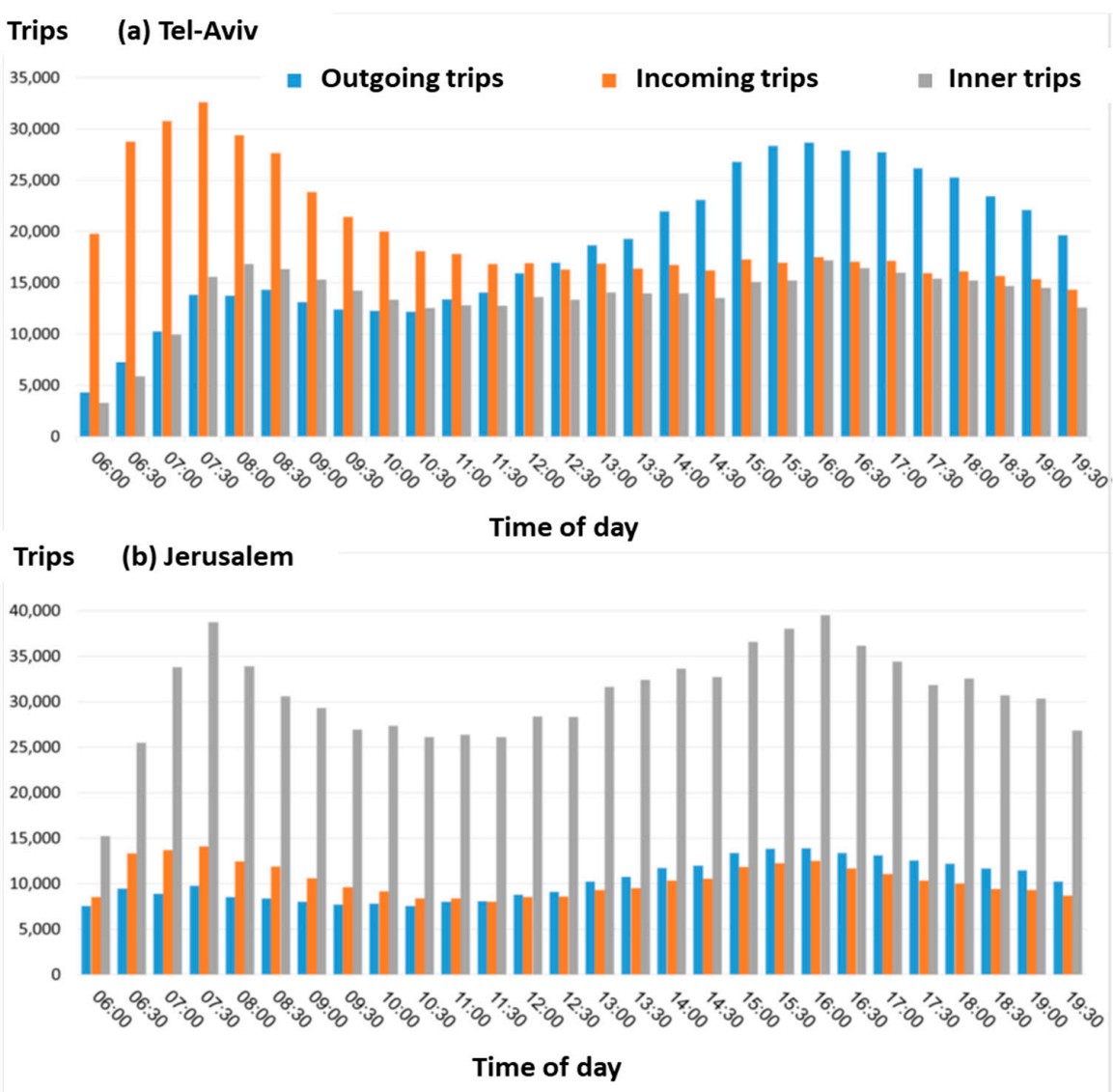

**Figure 5.** Ridership by half-hour distribution in (**a**) Tel Aviv and (**b**) Jerusalem.

### 4.2. Monthly Ridership Distribution

Data processing was performed based on the two years monitored, excluding the end of December 2019, due to a small quantity of ridership data for a given hour between specific O-D zones in a given month and considering privacy restrictions. Therefore, the analysis is based on cumulative data from 23 months. For the same reasons, detailed O-D tables are impossible to produce for a single month. However, more aggregate data can shed light on seasonal variation. Figure 6 shows vehicle kilometers traveled for each month divided by ranges of travel distance. Euclidean distances were converted to road distances and reflect a relatively small addition to mileage as the Euclidean distances become longer. Analysis was performed based on distances between the centroids of the zones. Ridership from the beginning of 2018 to the end of 2019 grew slightly, as shown in Figure 6.

The results also indicate that ridership properties seem to vary by season, as will be discussed in the commuting section comparing January (winter) to July (summer). The variation by season is also reflected in Figure 6, while analyzing ridership for each month, inflated to reflect the entire population. It is observed that July–August ridership is slightly lower compared to the other months but with longer mileage, reaching the peak of each year (which corresponds to monthly gasoline-consumption data).

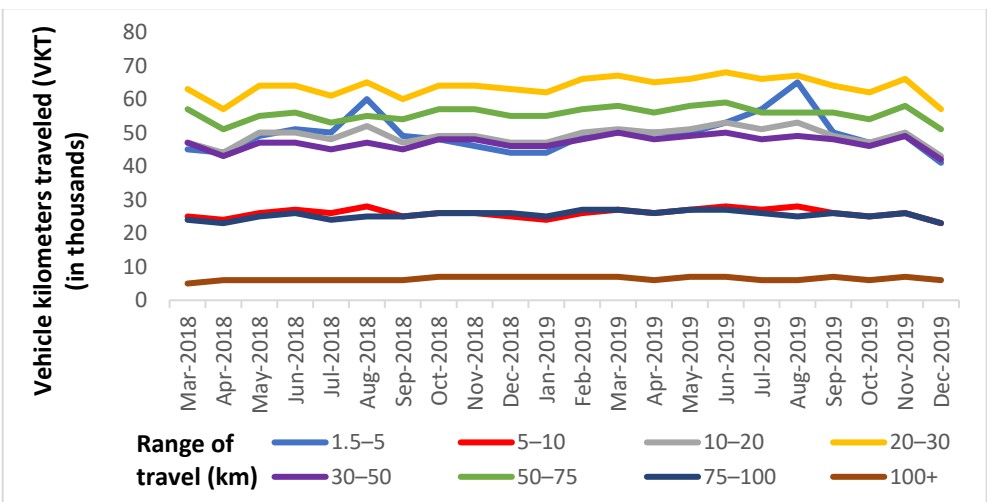

**Figure 6.** Vehicle kilometers traveled (VKT) by range of travel distance.

*4.3. Origin–Destination Trips*

The CD was processed at a level of 1270 O-D pairs, divided by day type and time of day. In total, 72 detailed O-D tables were generated, 24 for weekdays Sunday to Thursday and 24 for weekends. Figure 7a,b presents the distribution of average daily trips between the 15 superzones for workdays for the Israeli population and for tourists. The data were additionally produced for weekends and the 33 main zones. The data set allows for the production of detailed O-D tables reaching a level of detail for 1270 zones. The most prominent areas for incoming and outgoing rides are Jerusalem (approximately 1.5 million rides per day, out of which 1.1 million are internal rides) and Tel Aviv (1.12 million rides per day, out of which 0.46 are internal rides).

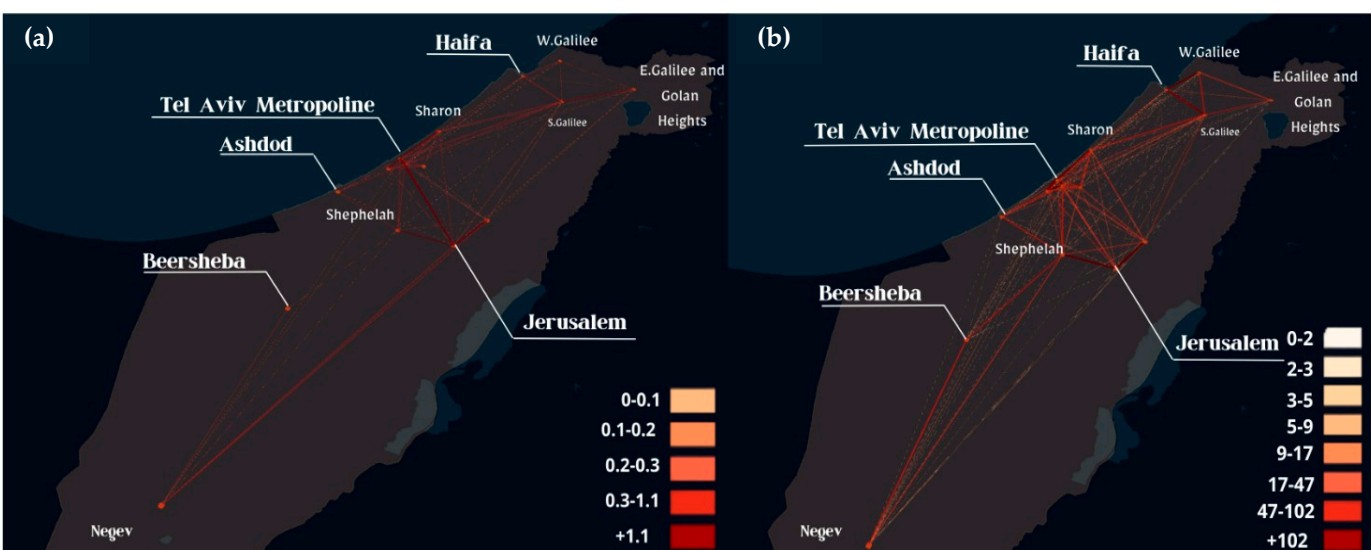

**Figure 7.** Average daily ridership on workdays by origin and destination (in thousands of km) for (**a**) tourists and (**b**) the Israeli population.

Figure 7b presents an average daily tourist trip between 15 superzones. These analyzed results are based on data from the Pelephone network alone, as Cellcom was not allowed to share tourist data due to legal restrictions. A total of 141,000 trips a day were identified and expanded on based on the ratio between the users from each country and the known number of incoming tourists from that country according to Israel's CBS. Overall, tourists added 0.99% to Israeli ridership. In terms of mileage, their contribution is slightly higher. Their most common destination is Jerusalem, followed by Tel Aviv. South and east of

Tel Aviv is prominent due to its inclusion of Ben-Gurion Airport. A main destination for tourists, the city of Eilat, is underrepresented, seemingly because of its direct airlines from Europe and little local travel of tourists between Israel's central region and Eilat.

Figure 8 presents incoming and outgoing ridership from each main zone during morning and afternoon hours; they are instructive in terms of each area's centrality, or they point to an area's dependence on other areas. Tel Aviv's centrality is clear in this aspect, based on the surplus of incoming rides in the mornings and the inverse surplus of outgoing rides in the afternoons. Jerusalem, Haifa, and Beer-Sheva attract high ridership as well. Petah Tikva has a minor surplus of incoming rides, while Holon-Bat-Yam, for example, seems to rely on other areas. The dominant explanation for this phenomenon is occupational dependence, although the analysis includes ridership for other purposes, e.g., education.

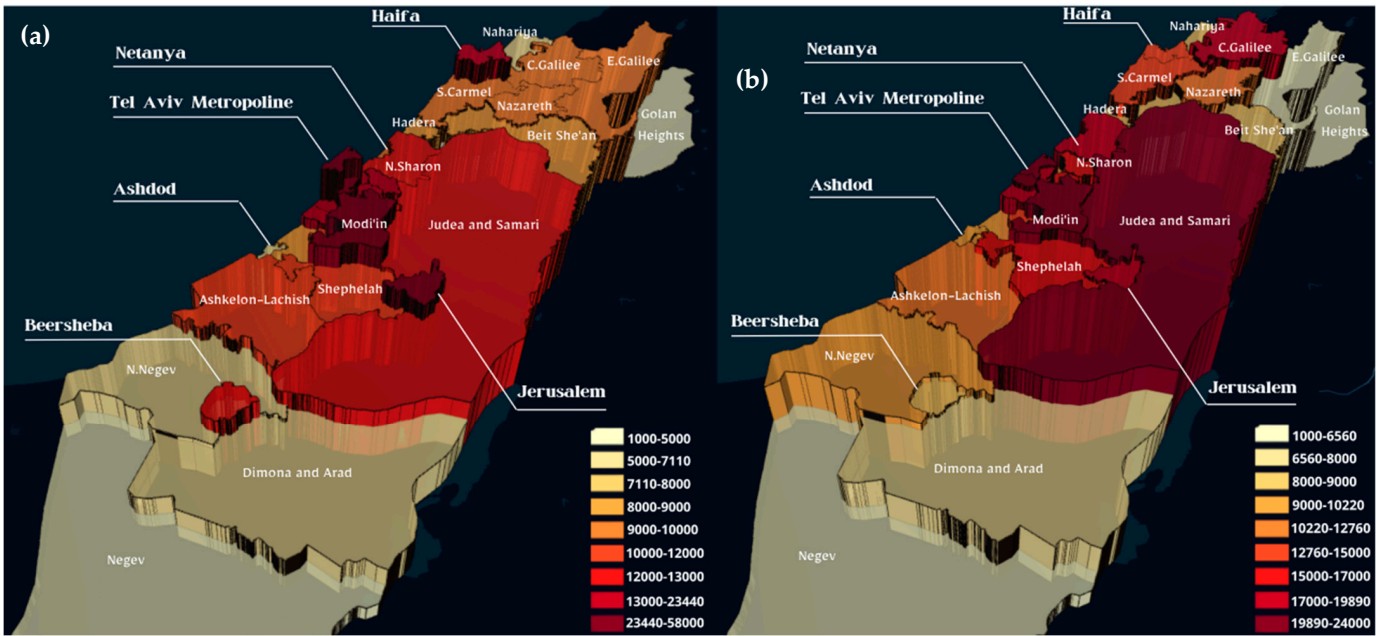

**Figure 8.** Ridership for central zones on weekdays at morning peak (6 am–9 am): (**a**) incoming; and (**b**) outgoing.

Additional matrices were produced for incoming rides to occupational areas/main commercial center areas, outgoing rides from different cities based on destination and period of the day, and ridership at the neighborhood/borough level for different cities. This section includes some examples from a large number of detailed processing outputs made possible thanks to the database acquired in this research.

### 4.4. Personal Data-Based Processing

Up to this point, processing was based on ridership observations between origin and destination zones but not on individuals' mobility patterns. Some processing was limited to the time and extent in which travel habits were examined at the level of users, but these were presented aggregately, in compliance with privacy restrictions. Figure 9 presents the number of recurring rides made on the same day and between the same O-D areas over the course of one month. The months January and July were chosen to capture different patterns of commuting, as the month of January represents a month with regular activity, while in July, students are on vacation. Outgoing rides are observed from a given area between 6 am and 10 am for a predetermined destination, including a stay of 3 h or more and a (direct or indirect) return to origin. This form of ridership, when frequent, is a commute, mostly to work but partially (depending on the season) for educational purposes as well (other travel purposes are possible). Following these assumptions, a person who travels to his work or school 5–8 times a month is defined as a

light commuter, and a person who travels 9–15 times a month or 16 or more times a month is defined as a medium commuter or heavy commuter, respectively. Figure 9 shows the proportions of light, medium and heavy commuters in Israel, based solely on Pelephone mobile provider data.

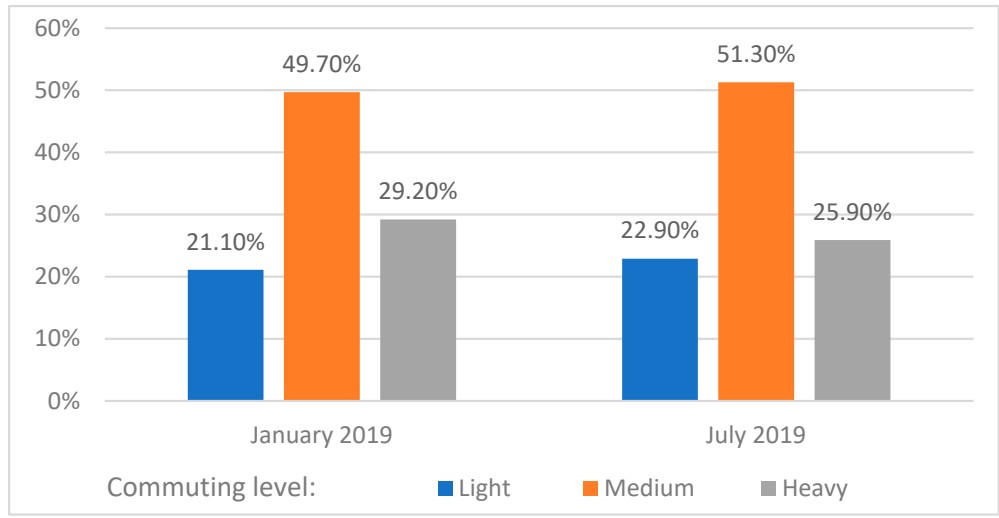

**Figure 9.** Light, medium, and heavy commuters in Israel.

The result is not sensitive to the change between typical work and holiday periods in Israel. It can be observed that in the holiday period, there are slightly fewer heavy commuters. This may be explained by the fact that the holiday period does not apply to employees but only to some university students and younger school students. Most school students travel to school within their area, and thus their travels or holidays do not affect commuting rates much.

*4.5. Validation*

This research included numerous validations, beginning with the experimentation phase (as described in the Methodology section) and up until the final outputs. These validations included ride logging, data comparison between the cellular operators and comparison with many external sources, such as GPS, Israel's CBS, THS, other surveys, and traffic monitoring. All these were performed across different time and area sectors. In this paper, selected validation outputs are presented.

Figure 10 presents the correlation between CD and the number of trips ending in cities located in Israel's four largest metropolitan areas, as measured in travel habit surveys taken in those metropolitan areas between 2014 and 2019 [46] (THS was collected in 2014 and 2016–2017 for Tel Aviv, in 2016–2017 for Haifa, in 2014–2015 and 2019 for Beer-Sheva, and in 2014–2017 for Jerusalem). As shown in Figure 10, the correlation is strong at a correlation coefficient of 0.98 throughout the day and a correlation coefficient of 0.99 during morning hours (6 am–11 am), with a smaller number of rides in the CD than in the travel habit surveys.

Figure 11a presents the hourly ridership distribution on workdays, excluding rides shorter than 1.5 km, as measured using the CD and compared to the THS. The results show that there is a strong correlation in regard to travel patterns by time of day, and the average difference between the surveys is 0.73%. The largest difference was found at 7 am and included a surplus of 4.4% trips according to the THS.

Figure 11b–e presents the same analysis divided into Israel's four largest metropolitan areas, when a consistent and extreme surplus of ridership is seen at approximately 7 am in the THS. This result is in line with Figure 10b, which points to a surplus in ridership in the THS during morning peak hours. While THS are a reliable source of comparison, they rely on one or two days of the sampled population, which are both assumed to be

representative. However, the analyzed CD averages two years of data of 50% of the Israeli population; thus, it is reasonable to assume that the cellular pattern is more reliable and that THS overestimates the number of trips in the morning peak.

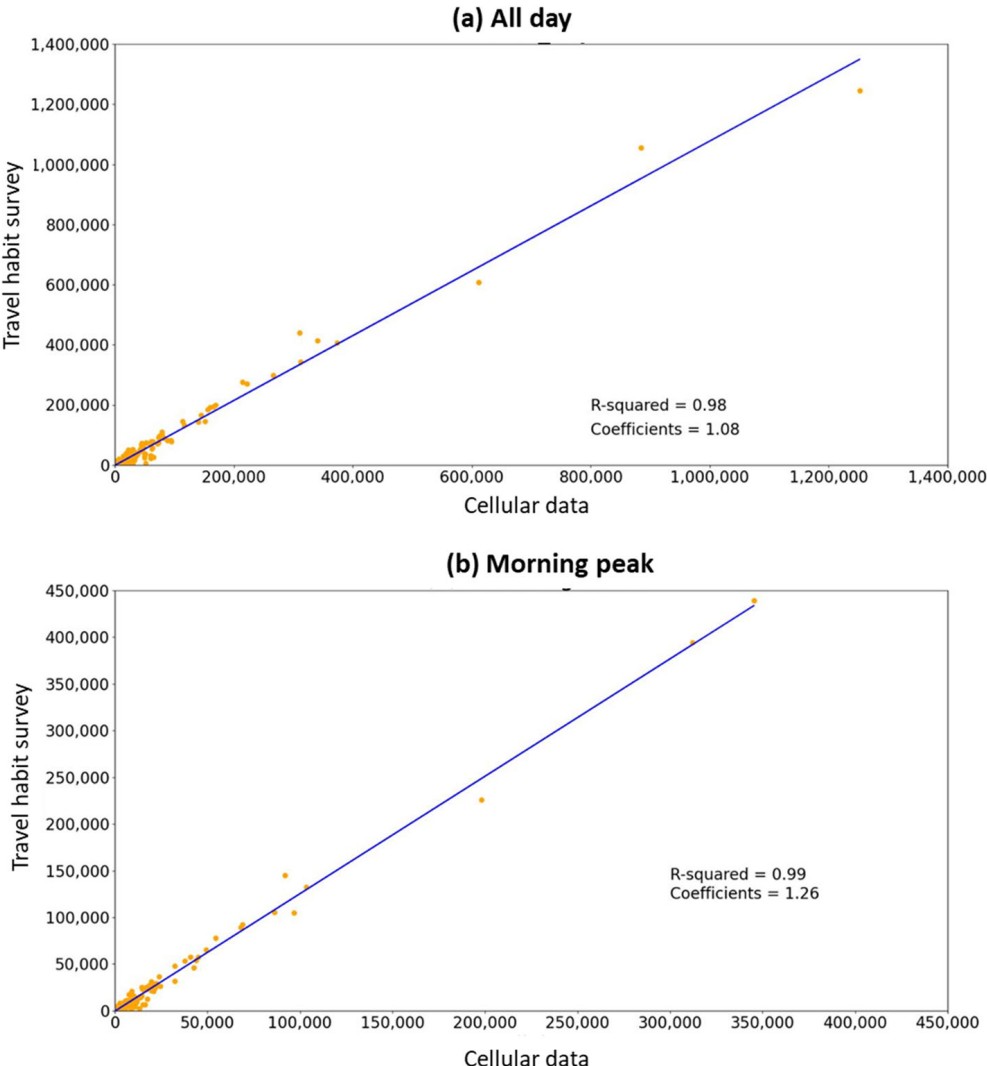

**Figure 10.** Trips to cities: correlation between the cellular data and travel habit survey for (**a**) all day and (**b**) morning peak hours.

Figure 12 analyzes the number of trips crossing three cordons (an area or a line network made up of a number of screenlines that completely enclose the specific area or district), A-B, B-C, and C-D, in the Tel Aviv metropolitan area, in both directors and in absolute number, as defined in Figure 12a. Figure 12b shows the number of trips crossing the three cordons A-B, B-C, and C-D in the Tel Aviv metropolitan area, in both director percentages, and Figure 12c shows the area (rings) A–D. The three databases of comparison are the CD, a cordon survey conducted in 2018, and the THS, conducted in 2017 in the Tel Aviv metropolitan area. This analysis includes all rides taken between 6 am and 8 pm, excluding rides shorter than 1.5 km and ridership of children under the age of 8. It can be seen that the percentage of rides that cross the cordons, in both directions, is similar across the three databases of comparison and that the CD and the cordon survey match the amount of rides that cross the cordons in absolute numbers. In the THS, there are fewer than 100k trips compared to the other data sources; however, the percentage of rides that cross the rings is similar. The lack of trips in the THS is because this source of data includes only private cars and taxis. In the other surveys, conversely, all types of vehicles were included.

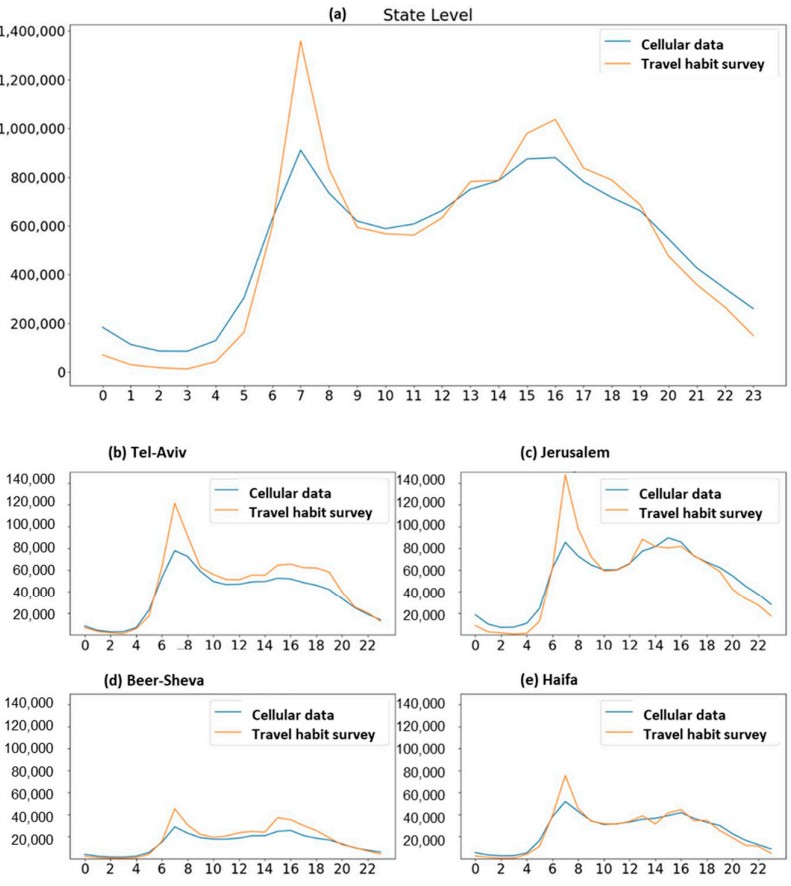

**Figure 11.** Trips to major metropolitan areas by time of day: a comparison between cellular data and travel habit survey at the (**a**) state level; (**b**) Tel Aviv; (**c**) Jerusalem; (**d**) Beer-Sheva; and (**e**) Haifa metropolitan level.

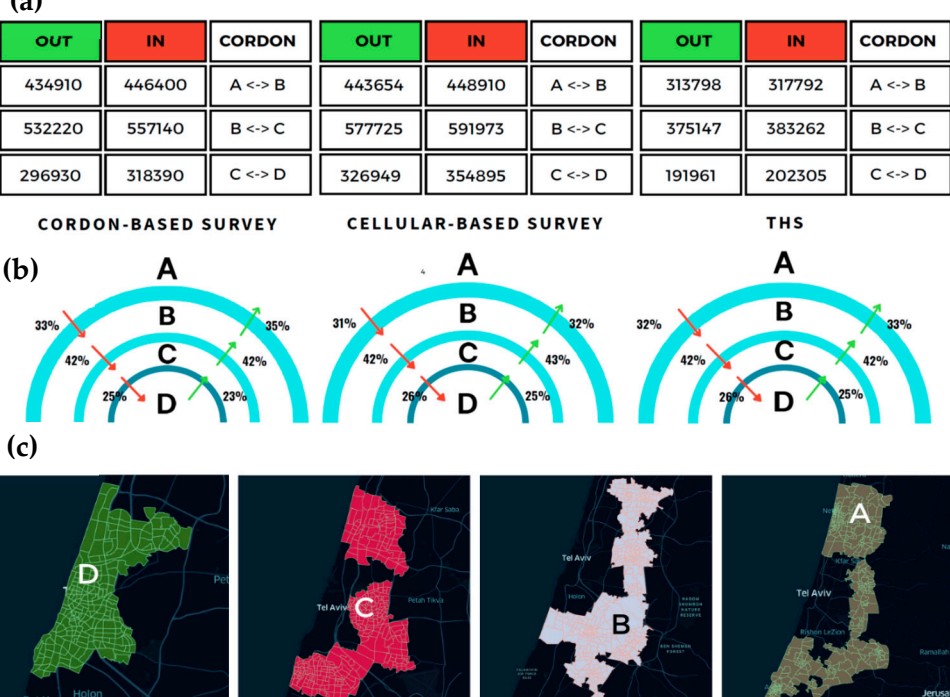

**Figure 12.** Trips crossing the Tel Aviv metropolitan area's cordons: A comparison between the CD, cordon survey and THS in (**a**) absolute numbers and (**b**) percentages. In (**c**), areas A–D are shown.

## 5. Discussion

This work focuses on one of the most extensive sets of CD of its kind carried out thus far in the world, which was performed for two years in 2018 and 2019 with the participation of the two largest cellular providers in Israel, as well as leading GPS companies. The large-scale cell phone data collection covered half the population aged 8+ in Israel and uncovered local and national trip patterns in a manner that reveals the structure of nationwide travel demand.

The data collection results included mainly O-D matrices varying in resolution, time section, and other transportation properties. These data were processed into basic outputs, which included O-D travel matrices, in different time sections and resolutions. Most of them are hourly matrices of 1270 by 1270 zones (by weekdays and weekends). These include half-hour matrices or daily matrices with a multiplied resolution of 2640 by 2640 zones. Additionally, extended outputs were produced, which included trip matrices sectioned by home–work association, commuting indicator, travel frequency, tour duration, and mobility of tourists. In this paper, only selected results are shown.

Numerous validations were performed, beginning with the experimentation phase and up until the final outputs. These validations included ride logging, data comparison between the cellular operators and comparison with many external sources, such as GPS, Israel's CBS, THSs, traffic monitoring surveys, etc. All these were performed across different time and area sectors. The long-term collection of the data, the extensive coverage of the population, the close control over the analysis, the participation of various operators, and the methodological solutions developed allowed for the extraction of results with high accuracy and representativeness. Indeed, the results indicate a strong correlation between the CD and other reliable sources. Such a result suggests that using CD can replace trip generations/attractions and the origin–destination matrix generation step as part of the four-step model procedure and, more importantly, allow us to generate such data regularly. Furthermore, the O-D matrices obtained from cellular data can directly be transferred to the third step out of four stages of the four-step model, which includes the allocation of the matrix to different modes and finally the assignment of the trips.

The use of such large-scale CD may enrich the applications in the field of demand modeling and planning in a way that was thus far impossible. It may be used for updating the demand for a national model while also expanding and maintaining such a model. It can be used to support the design and management of public transport systems by processing the data for public transport planning usage. Furthermore, it can be used for updating and validating local and metropolitan models for the purpose of transportation project evaluation. It can also be used for demand management in real time, ongoing monitoring, and more. Equally important is the ability to develop, using the collected data, transportation measures at the regional and local levels, such as inner trip matrices, trip attraction and generation, trip distance, travel times, percentage of use of public transportation, intraregional trips vs. trips to other regions, etc., for better planning of projects and planning processes by providing relevant and up-to-date data.

The information collected in this work is very extensive, detailed, and rich, yet its use requires caution. At the level of a 1270-zone breakdown, there may be a "spillover" of trips from one region to another. There are zones with an area of 500 hectares or even less (700 m square), while the location accuracy limits of antennas are 500 m on average; thus, a trip attributed to a given zone can actually be associated with a bordering zone. Errors can be reduced by considering two adjacent zones. Furthermore, the relatively low extent of the detection rate for trips shorter than 1.5 km does not allow for an in-depth understanding of short distance movement, and such technology limitations are expected to improve with the introduction of "Generation 5" of mobile phones to the market.

The data collected and analyzed in this work are relatively basic, and there is room for many future in-depth studies. Future work should explore the validation and completion of the methodology, mainly in the field of traveler characteristics and modes of travel monitoring. More specifically, the ability to infer the mode of travel from CD should be investigated.

In addition, this includes the ability and method to integrate available information such as population and land use data with CD for the purpose of expanding the uses of CD for transport planning. Furthermore, the ability to map and monitor the movements of vehicles and populations based on CD may also be an interesting and attractive direction for other areas, such as business, tourism, health, security, and education.

**Author Contributions:** The authors confirm contributions to the paper as follows: Data curation: Vladimir Simon study conception and design: Shuki Cohen, Israel Feldman and Bat-hen Nahmias-Biran; data collection and method development: Shuki Cohen and Israel Feldman; analysis and interpretation of results: Bat-hen Nahmias-Biran, Shuki Cohen and Israel Feldman; manuscript preparation: Bat-hen Nahmias-Biran and Israel Feldman. All authors have read and agreed to the published version of the manuscript.

**Funding:** This work was supported through the Research and Design of Economic Policy Brunch in MoT, Israel.

**Institutional Review Board Statement:** Not applicable.

**Informed Consent Statement:** Not applicable.

**Data Availability Statement:** The full data set used for this study is publicly available. The cellular data can be found in [5]. Data exploited for validation may be found in [46].

**Acknowledgments:** The authors would like to acknowledge the contribution of, Leonid Heifetz, and Hillel Bar-Gera. The partial institutional support of Ariel University is much appreciated.

**Conflicts of Interest:** The authors declare no conflict of interest.

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
