# Peer review of "Large-Scale Mobile-Based Analysis for National Travel Demand Modeling"

_ijgi, doi:10.3390/ijgi12090369_

Round 1

Reviewer 1 Report (Previous Reviewer 2)

No comments.

Author Response

Thank you for your time and efforts. No comments were submitted and therefore no corrections were made.

Reviewer 2 Report (New Reviewer)

Statistical  validation is not carried out

Source of the data is not known and no reference eg. Figure 3,4,5

Accuracy test of origin-destination matrixes at different resolutions, 23 revisions of algorithms and reproduction of data

 Travel Habits Surveys (THS)). Reference

 comparison between the output of  140 volunteers data and CD

Very difficult to follow need  major  revision write up

Author Response

Reviewer 2:

Statistical  validation is not carried out

RE: Statistical validation is carried out. Please see Validation section, and specifically Figure 10 which presents the correlation between CD and the number of trips ending in cities located in Israel’s four largest metropolitan areas, as measured in travel habit surveys taken in those metropolitan areas between 2014 and 2019. The statistical analysis is presented in terms of correlation coefficient and R^2.

Source of the data is not known and no reference eg. Figure 3,4,5

RE: Source of the data of Figure 3,4,5 is the cellular data. Clarifications were added throughout the text. Reference was indeed missing and was added accordingly (reference no. 5).

Accuracy test of origin-destination matrixes at different resolutions, revisions of algorithms and reproduction of data

RE: Accuracy test of origin-destination matrixes at different resolutions was done as much as possible. For example, we present such a test for destinations in figure 10 where the correlation between CD and the number of trips ending in cities as measured by the THS is calculated. Such calculation is also presented in figure 11 while dividing into metropolitan areas. The statistical differences are discussed in validation section (section 2 and 3). Accuracy test was also performed in comparison to traffic counts (see figure 12). Furthermore, accuracy test for preliminary study was added and presents location accuracy as measure by GPS providers compared to CD providers.

Accuracy tests were also performed for corridors, comparison with train passengers’ data etc. Most of them are discussed in: https://www.matat.com/_files/ugd/eff2a0_94c12c34791f4ceda09ef3ee44849094.pdf
(in Hebrew however).

Travel Habits Surveys (THS)). Reference

RE: Thank you. Reference was added.

 comparison between the output of  140 volunteers data and CD

RE: Thank you for this suggestion. Comparison between the output of 140 volunteers’ data and CD was added as a new table – Table 2.

Reviewer 3 Report (New Reviewer)

Dear Authors,

The paper is very well structured. The content and theme of the article is consistent with the lines of the journal and the topic is of interest to the readers.

Overall the presentation is very good. However, the paper has some inconsistent compering to the instructors for the authors, which should be corrected:

·        The title of the manuscript are concise, specific and relevant. This is Ok.

·         The Abstract contain around 200 words. This is Ok.

·     The Abstract contain all main obligatory elements (according to the instruction to the authors): Background of the research, Methods; Results; Conclusion. This is Ok.

·         List of Keywords is appropriate.

·   The paper contain most of main obligatory chapters (Introduction; Materials and Methods; Results). This is Ok. However, the Discussion chapter is missing. It is  integrated with Conclusion chapter and present results of the research. This is Ok.

·        Introduction chapter contain all mandatory elements such are: define purpose of the work, defining specific hypotheses which have being tested, current state if the research field, key publication from the filed cited. However, main conclusion are not presented. Should be added.

·      In the paper is often used expression “study”, “in this study” or “for this experimental study”. However, instead of study authors should use expressions like “research” or “in this paper” etc.

·         The paper should be written in third face of singular. So first face of singular is not appropriate (should not use expression “we”, “our” etc.). Should be corrected.

·       Data and Methods chapter in detail describe provided research with sufficient information for replication of provided research.

·     The Result chapter provide concise and precise description of the experiment results.

·       The references are numbered in order of appearance in the text in the text. This is Ok. However, the references must be presented in the text in the square brackets and not regular brackets. Should be corrected.

·         The paper has only 17 references, which is unusually small number of references. Should be extended.

·         In the paper are presented equations which are numbered in brackets. However, those brackets are not placed on the right margin of the text. This should be corrected.

·         It is recommendation that authors use same letter font in tables as is in text of the paper.

Author Response

Reviewer 3:

Dear Authors,

The paper is very well structured. The content and theme of the article is consistent with the lines of the journal and the topic is of interest to the readers.

Overall the presentation is very good. However, the paper has some inconsistent compering to the instructors for the authors, which should be corrected:

  •       The title of the manuscript are concise, specific and relevant. This is Ok.
  • The Abstract contain around 200 words. This is Ok.
  •    The Abstract contain all main obligatory elements (according to the instruction to the authors): Background of the research, Methods; Results; Conclusion. This is Ok.
  • List of Keywords is appropriate.
  •  The paper contain most of main obligatory chapters (Introduction; Materials and Methods; Results). This is Ok. However, the Discussion chapter is missing. It is integrated with Conclusion chapter and present results of the research. This is Ok.

RE: Discussion chapter was added instead of conclusion section as per Editor suggestion.

  • Introduction chapter contain all mandatory elements such are: define purpose of the work, defining specific hypotheses which have being tested, current state if the research field, key publication from the filed cited. However, main conclusion are not presented. Should be added.

RE: Main conclusions were added to the Introduction chapter as follows:
“Results indicate a strong correlation between the CD and other reliable external sources such as GPS, Israel’s CBS, THSs, and traffic monitoring surveys.”

  •     In the paper is often used expression “study”, “in this study” or “for this experimental study”. However, instead of study authors should use expressions like “research” or “in this paper” etc.

RE: The expression “study” was replaced with expressions like “research” or “in this paper” as suggested.

  • The paper should be written in third face of singular. So first face of singular is not appropriate (should not use expression “we”, “our” etc.). Should be corrected.

RE: The paper is now written in third face of singular.

  •      Data and Methods chapter in detail describe provided research with sufficient information for replication of provided research.
  •    The Result chapter provide concise and precise description of the experiment results.
  •      The references are numbered in order of appearance in the text in the text. This is Ok. However, the references must be presented in the text in the square brackets and not regular brackets. Should be corrected.

RE: The references are now presented in the text in square brackets and not in regular brackets.

  • The paper has only 17 references, which is unusually small number of references. Should be extended.

RE: A new section “Related Work” was added along with a significant number of references.

  • In the paper are presented equations which are numbered in brackets. However, those brackets are not placed on the right margin of the text. This should be corrected.

RE: Equation numbering has been fixed, and now appears on the right margin of the text.

  • It is recommendation that authors use same letter font in tables as is in text of the paper.

RE: Tables font is now the same as text font.

Round 2

Reviewer 2 Report (New Reviewer)

No comments

Author Response

We thank the reviewer for his/her time and efforts. In this round, no comments were given and therefore no corrections were made.

This manuscript is a resubmission of an earlier submission. The following is a list of the peer review reports and author responses from that submission.

Round 1

Reviewer 1 Report

This paper devotes to the national travel demand modeling with large-scale mobile data, where lots of investigations have been designed. It is a topic of interest to the researchers in the related areas but a number of points need clarifying and certain statements require further justification. There are given below.

1. It is necessary to elaborate or give the basis for identifying the trip in section 2.2.1.

2. This paper introduces the Expansion factors to generate the O-D matrixes for the full population. It is difficult to confirm the representations in expanding data samples. Can you give more explanations or proof of experiments?

3. There are many imprecise details in the paper, such as:

1) Page 10 Fig.3 has some incorrect box labels.

2)The vertical axis of Page 13 Fig. 6 is unclear.

3) Please carefully recheck the Fig in the paper.

Therefore, I recommend a major revision of this paper.

Reviewer 2 Report

- The paper lacks a detailed section on “Related work”. Few papers are mentioned in the Introduction about using mobile phones data, but most important is to present related work about Big Data collection/gathering about mobility.

- It is not clear the objective of the paper. I understand the analyses are data-driven, but even so a kind of “research question” must exist otherwise we do not know why the analysis are being made and what we are expecting to find.

-Details about the data and the methods should not be included in the Introduction, but in the appropriate section: Data and Methods.

- All the acronyms must be presented comprehensively at least once: CD, O-D, CBS, …

- It is not correct to use the term “survey” to describe the data used. A survey is a process of collecting primary data by administering a questionnaire to a sample of respondents. This was not the process entailed to gather the data.

- Statistics about penetration rate of mobile phones – and more precisely smartphones – in Israel should be provided.

- The paper is opaque in methodological details. For example, line 149 “Such methods undergone several rounds of improvements, calibrations and optimizations”. Another example, line 219 “(..) and the algorithms that associate location data with the cellular phone”. What exactly was done?

- No explanation is provided to set “8 minutes” and “40 minutes” (line 170-173) as “thresholds” for stop length; In Figure 1 there is a trip time of 16. Why were these values chosen?

- Only travels “work” and “home” were considered. How did the authors deal with travels to the supermarket, to the doctor, to visit a relative, …?

- What is the meaning of “expansion factors”? Extrapolation?

- Figure 2: X axis and Y axis should have the same scale

- Figure 3 is confusing, it has a table overlapping; what do the colours mean in this figure?

- Please check lines 340-341. The interpretation is not consistent with Figure 4a.

- Figure 4a and 4b should represent “time of the day” as a.m. and p.m.  The same on Figure 5 and Figure 11

- Figure 4b – what is being represented, the time when the trip started or ended?

- Line 392 – only data from “Pelephone” were analysed. What about the other operator?

- Figure 7 and Figure 8 – where is (a) and (b)

- Section 3.4 needs revision – there is confusion between Figure 9 and Figure10.

- Figure 9 – why were January and July the chosen months to represent outcomes?

- Line 433 – what is the reasoning to define 5-8 minutes as “light”, 9-15 as “medium” and “16+” heavy?

- Line 449 – “between 2014 and 2019” – wasn’t the CD data referenced to the period 2018-2019?

- Figure 11 – legend must detail what is (b), (c), (d) and (e)

- What is the relevance of Figure 11? If the authors argue in advance that CD data is “naturally” better because is the average of two years, did the authors expect to conclude differently with Figure 11? (lines 476-479)

- Figure 12 is confusing – is like three figures in one and it is scarcely explained in the text.

- Line 534 “Indeed, results indicate a strong correlation between the CD and other reliable sources”. Is this an enthusiastic result? If CD and other sources allow to conclude the same, what is the advantage of using CD?

- In substantive terms: did the authors find out something about mobility patterns in Israel they did not know already from other sources? It seems the authors are validating a new methodology to study travel patterns. If that is the case, I suggest publishing in a Big Data Methods journal instead of Geo-Information Journal.

- A lot of typos throughout the text – a full revision is required.

Reviewer 3 Report

This is a well-written paper that covers an important topic and contains a sound methodology. I have a few concerns about the paper as the following:

*)  Obviously, the methodology presented in the paper is strongly related to the scope of the paper. Nevertheless, I think that a few paragraphs can be added to the text to emphasize the usefulness of obtaining the O-D matrices with the use of cellular data from the technical point of view. Actually, in the Introduction section, this is pointed out albeit in a limited manner and a relatively non-technical text. Technically, a classical four-step model yields the O-D matrices in two steps; first the trip generations/attractions are modeled and next the O-D matrices are acquired according to these generations/attractions and based on travel impedances. The methodology developed in this paper, removes these two steps and more importantly, provides data that can be updated regularly. By this way, the need for estimating regression models for generations/attractions as well as the calibration of gravity-type models are omitted. Mentioning these aspects in a technical manner would definitely improve the paper's reach to wider audiences. I also think that the authors can include this technical summary mostly by referring to reference no. 1, Modelling Transport, which is a seminal book in transport modelling. In addition, maybe, the remaining steps to complete a standard demand model can be explained with a few sentences as well (i.e., the O-D matrices obtained from cellular data can directly be transferred to the mode choice step, etc.)

*) Parallel to the point I mentioned above, I think that the use of cellular data can further be extended to determine mode choices by applying proper methodologies. On page 18, row 545, the authors mentioned % use of public transportation as a measure that can be determined. This is actually (a part of) the modal split step of the 4-step model. If this % can be determined, maybe the %s of other modes can be determined as well. And maybe, this might lead to three steps of the 4-step model to be completed by using cellular data.  I believe, this can be specifically stated as a future study in this paper.

*) On Page 4, the authors talked about volunteers that installed a specific app on their phones. I think it would be informative the give the number of volunteers.